# Influence of Green Synthesized Zinc Oxide Nanoparticles on Molecular Interaction and Comparative Binding of Azure Dye with Chymotrypsin: Novel Nano-Conjugate for Cancer Phototherapy

**DOI:** 10.3390/pharmaceutics15010074

**Published:** 2022-12-26

**Authors:** Amit Singh, Pankaj Kumar, Niloy Sarkar, Mahima Kaushik

**Affiliations:** 1Nano-Bioconjugate Chemistry Lab, Cluster Innovation Centre, University of Delhi, Delhi 110007, India; 2Department of Chemistry, University of Delhi, Delhi 110007, India; 3Department of Environmental Studies, University of Delhi, Delhi 110007, India

**Keywords:** nanoparticles, Zinc oxide nanoparticles, A-549 cell line, AzureC, chymotrypsin, cancer phototherapy, dye-ZnO complex, protein-nanoparticle interaction

## Abstract

Till date, different types of conventional drugs have been used to fight tumors. However, they have significant flaws, including their usage being constrained because of their low bioavailability, poor supply, and serious side effects. The modern combination therapy has been viewed as a potent strategy for treating serious illnesses, including cancer-type feared diseases. The nanoparticles are a promising choice for cancer therapeutic and diagnostic applications because of their fascinating optoelectronic and physicochemical features. Among the metallic nanoparticles, Zinc oxide nanoparticles possess interesting physicochemical and anti-cancer characteristics, such as ROS generation, high retention, enhanced permeability etc., making them attractive candidates for the treatment and diagnosis of cancer. Zinc oxide nanoparticles showed anti-cancer property via excessive reactive oxygen species (ROS) production, and by the destruction of mitochondrial membrane. Here, we have synthesized organic/inorganic hybrid nanosystem composed of chymotrypsin protein (Chymo) with AzureC (AzC) conjugated with Zinc oxide nanoparticles (ZnONPs). The conjugation of AzureC with ZnONPs was confirmed by transmission electron microscopy (TEM), zeta potential, and dynamic light scattering (DLS) experiment. The interaction of Chymo with AzC alone and AzC-ZnONPs was investigated, and it was observed that the interaction was enhanced in the presence of ZnONPs, which was concluded by the results obtained from different spectroscopic techniques such as UV-Visible spectroscopy, fluorescence spectroscopy and circular dichroism in combination with molecular docking. UV-Visible spectroscopic studies and the corresponding binding parameters showed that the binding of AzC-ZnONPs complex with Chymo is much higher than that of AzC alone. Moreover, the fluorescence measurement showed enhancement in static quenching during titration of Chymo with AzC-ZnONPs as compared to dye alone. In addition to this, circular dichroism results show that the dye and dye-NPs conjugate do not cause much structural change in α-Chymo. The molecular docking and thermodynamic studies showed the predominance of hydrogen bonding, Van der Waal force, and hydrophobic forces during the interactions. After correlation of all the data, interaction of Chymo with AzC-ZnONPs complex showed strong interaction as compared to dye alone. The moderate binding with chymo without any alteration in the structure makes it desirable for the distribution and pharmacokinetics. In addition, the in vitro cytotoxicity of the AzC-ZnONPs was demonstrated on A-549 adenocarcinoma cell line. Our findings from physiochemical investigations suggested that the chymotrypsin coated AzC conjugated ZnONPs could be used as the novel nanoconjugates for various cancer phototherapies.

## 1. Introduction

Cancer continues to be one of the leading causes of mortality worldwide. According to statistics, more than 19 million cases were reported throughout the world till September 2022 [1], with estimated 10 million deaths by 2020 [1,2]. The scientific community is continuously working to create new diagnostic tools and therapeutic approaches to deal with this deadly disease. However, several issues, including drug resistance and non-targeted activity, restrict chemotherapy’s effectiveness and produce undesirable side effects. A potential strategy to improve anticancer therapy is to reduce unfavorable systemic toxicity, as well as to minimize side effects in multimodal therapy, which results in a synergistic or combined impact of two separate therapeutic modalities [3].

The cationic phenothiazinium dyes, such as methylene blue, toluidine blue O, phenosafranin (PSF), safranin-T (ST), safranin-O (SO), and azure dyes etc., are used for the various types of cancer therapies, designing biosensors and DNA, RNA binding purposes [4,5,6,7,8,9,10]. Among them, AzureC is a water-soluble thiazine dye having important photosensitizing properties, which can induce photoactivated dye toxicity in a DNA nicking assay [11]. The therapeutic use of the AzureC and its photodynamic property (PDT) is very limited due to the lack of efficacy and system administration. In some cases, the derivatives witness the inactivation of the chemical structure in the biological environment [12]. These dyes alone have a few limitations, such as low solubility, stability, lack of specificity, etc., which affect their therapeutic efficiency. Modern treatment methods require a combination of therapies for the effective killing of cancerous cells and to reduce multidrug resistance. Researchers are now making significant efforts for the development of a nano assembly, which can solve the issues of dye solubility along with stability and enhance its efficacy.

In the past few years, nanomaterials have shown amazing potential as a drug and gene carrier. These nanomaterials have unique physical and chemical properties such as magnetic, optical, and mechanical properties, etc., which attract the attention of various industries, e.g., cosmetics, agriculture, electronics, and pharmaceutics, etc. [13]. The recent developments in nanoparticles are emerging as a revolutionary material in the biological and therapeutic process, e.g., drug and gene delivery, cell pathway, biosensing, bioimaging, etc., for the better cure of diseases. Zinc oxide nanoparticles (ZnONPs) have emerged as a successful material for gas sensing, piezoelectric, optical purpose, and photothermal property, etc. [14]. The ZnONPs have high catalytic efficiency, less toxicity, more biocompatibility, high adsorption tendency, and electron transfer kinetics, which make them a suitable carrier. On the basis of synthesis techniques, ZnONPs offer a variety of shapes, sizes, and provide a large surface area for the bioconjugation of ligands, dyes, and drugs, etc. [15]. Among the various synthesis techniques, green synthesis gained a lot of attention due to various advantages over chemical-based synthesis such as non-toxic side products and cost-effective green route of synthesis. [16]. The presence of various flavonoids, and terpenoids, etc. in plant extracts work as a strong reducing as well as capping agent without any harmful side products [17]. The features of metal or metal oxide-based nanoparticles and their interaction with biomolecules encourage their use as a delivery vehicle [18]. The use of nanocarrier for the delivery of dye for therapeutic purposes enhances the targeted and localized effects with low toxicity, non-invasive phototherapies and reactive oxygen species (ROS) generation to stimulate the cell apoptosis [8]. The uses ofdye-conjugated nanocarriers have recently attracted a lot of attention for the treatment of cancer, which can help in its delivery directly to the target site and may also increase its efficacy [8].

The recent development in bio-nanotechnologies utilizes the mutual properties of protein–nanoparticle conjugates, which can help in the advancement in various fields such as photothermal treatment, drug administration, imaging, biocatalysis, and biosensing [19]. Following the adsorption of protein onto nanoparticles, various proteins, including chymotrypsin, lysozyme, fibrinogen, and RNase A, have been found to significantly lose their function [20,21,22,23,24]. A detailed knowledge of the protein–nanoparticles conjugate is essential to know the effect of the structural changes in protein after interaction with nanoparticles. When nanoparticles enter the body and interact with protein, the protein binds selectively on the surface of nanoparticles (depending on the surface property) and forms a coat called protein corona [25]. Various studies have reported the stable binding of ZnONPs with proteins, e.g., the binding of transferrin protein and BSA with ZnONPs [20,21,26]. The adsorption of protein on the surface of nanoparticles forms nanoparticle-protein complexes which are known as nanoparticle-protein corona (NP-PC). This complex formation enhances the biocompatibility and biological activity of the complex. [22,25].

Proteins play a vital role in almost all the cellular activities and their thermodynamically stable folded structure under physiological condition is very important to perform different functions. α-Chymotrypsin (α-Chymo) belongs to the S1 family of enzymes and in the human body, the pancreas produces this digesting proteolytic enzyme chymotrypsin, which is employed in the small intestine to aid in protein digestion. Since the 1960s, the enzyme has been utilized in clinical healthcare settings and is also employed to aid in the creation of medications. It is well recognized to aid in promoting quicker healing of wounds and severe harm to tissue as well as assisting in reducing redness and swelling brought on by surgery or infection [27]. The adaptable enzyme is frequently included in pharmaceuticals to aid patients with sinus infections, bronchitis, or asthma. Chymo also plays a major role in the cleavage of peptides, digestion, blood clotting and blood pressure control. The α-Chymo is a globular protein, which contains three chains of 245 amino acids, which are connected by five disulfide bonds [27]. The structure contains a Greek key motif like two anti-parallel beta barrel domains followed by an anti-parallel hairpin [28]. It is a multifunctional protein and can bind with various small ligands and transport them to various target sites. Therefore, it is important to understand the comparative interaction between Chymo protein with AzC and nanoparticle-dye complexes to investigate this study further in the bloodstream to know its pharmacokinetics and pharmacodynamics. In a few reported studies, azure dyes bind strongly with the RNA, DNA, and proteins [29], hence, it’s binding with protein and NPs carriers helps in its adsorption, transport, and activity in the body. The revelation of various binding parameters helps in the use of Chymo-AzC-ZnONPs complex in various drug designing. The Chymo-AzC-ZnONPs complex can modify the therapeutic efficiency and potential of the AzC.

In this paper, we have reported the comparative interaction of Chymo protein with AzureC dye and AzC-ZnONPs using various techniques, such as UV-Visible absorption spectroscopy, fluorescence spectroscopy, circular dichroism, FTIR, and molecular docking, to understand the effect of ZnONPs on the potential binding of AzC with Chymo. Moreover, the cytotoxicity of AzC-ZnONPs on human lung adenocarcinoma cell line (A-549) was also performed using the MTT assay to assess their anti-cancer potential.

## 2. Material

α-chymotrypsin (α-Chymo) extracted from bovine pancreas (C4129 Sigma-Aldrich, St. Louis, MI, USA, Type II, Lyophilized powder, ≥40 units/mg protein), disodium salt of ethylene diamine tetra acetic acid (EDTA), AzureC (AzC) were purchased from Sigma. Zinc acetate dihydrate, sodium hydroxide, and sodium chloride were purchased from SRL India. All the experiments were performed in phosphate buffer (pH 7.4). The concentration of AzCwas determined by UV-Visible spectroscopy using the molar extinction coefficient of AzC as 73,000 M^−1^ cm^−1^ at 617 nm [30]. All the experiments were performed in double distilled water.

For the cell culture experiments, A-549 cancer cell was procured from NCCS Pune, in addition to fetal bovine serum (Gibco India), Dulbecco modified Eagle medium (DMEM), 96-well plates (ThermoFisher scientific), trypsin-EDTA solution (Gibco), antibiotic-antimycotic (100X), MiliQ water, and 3-(4,5-dimethylthiazol-2-yl)-2,5-diphenyltetrazolium bromide (MTT). All of the chemicals used in the tests are one of the highest quality available on the market.

## 3. Methods

### 3.1. Synthesis of Zinc Oxide Nanoparticles

The green synthesis of Zinc oxide nanoparticles was carried out by our earlier reported method by Singh et al. [16]. The green synthesized ZnONPs were characterized by UV-Visible spectroscopy, SEM, FESEM and XRD as presented in our previously published work [16]. The prepared ZnONPs were spherical in shape with a size of around 40 nm and gave a sharp UV-Visible absorbance peak at 368 nm. The synthesis of AzC conjugated green synthesized ZnONPs (AzC-ZnONPs) was performed using the technique reported by Sonia et al. with further modification [8]. Briefly, earlier green synthesized ZnONPs (1.0 mM) were taken in a centrifuge tube and then drop wise AzC (12.0 μM) was added to the ZnONPs suspension and incubated overnight. After incubation, the AzC-ZnONPs were centrifuged for 5 min and the obtained supernatant was discarded to get the pallet. Later, the pellet was washed to remove the excess dye, and then it was dissolved in the buffer for the experiments.

### 3.2. UV-Visible Experiments

UV-Visible spectroscopic technique was used to confirm the synthesis of nanoparticles and to carry out the interaction studies. UV-Visible spectra were recorded on Shimadzu UV–Visible spectrophotometer (UV-1650 PC). All spectra were taken in the wavelength range of 200–700 nm in a quartz cuvette of 1.0 cm path length. The stock sample of Chymo was prepared by using the Beer–Lambert law and taking the molar extinction coefficient value at 280 nm. Prior to the interaction studies, the ZnoNPs were incubated with AzC to form a complex (AzC-ZnONPs). Then, the interaction studies were performed by keeping constant concentration of Chymo (2.0 uM) and varying the amount of AzC or AzC-ZnONPs complex. Further, reverse titration was carried out by taking constant amount of AzC (12.0 uM) or AzC-ZnONPs complex (12.0 uM) and varying the concentration of Chymo from 0.8–2.0 µM. All the titrations were performed in sodium phosphate buffer of pH 7.4.

### 3.3. Fluorescence Experiments

To understand the binding nature of chymotrypsin, fluorescence quenching experiments are used quite often. The fluorescence measurements were conducted on Hitachi F-4700 spectrofluorometer equipped with a temperature controller using 1.0 cm path length fluorescence cuvette. The Chymo molecule was taken as a fluorescent probe to study the interaction with AzC dye of different concentrations in the presence and absence of ZnONPs (1.0 mM). The excitation wavelength was taken at 295 nm to selectively excite the tryptophan unit at fixed slit width of 5.0 nm. The samples were prepared in phosphate buffer at pH 7.4 and the spectra were taken in the range of 300–450 nm wavelength at two different temperatures, 298 K and 323 K.

### 3.4. Circular Dichroism Experiments

The circular dichroism (CD) experiments were performed on JASCO J-815 spectrometer with constant nitrogen purging. The CD spectra of chymotrypsin were recorded with different concentrations of AzC alone and AzC-ZnONPs to monitor their effect on native chymotrypsin structure. The samples were prepared in phosphate buffer (pH 7.4) and the average of three successive scans was taken while operating in the range of 190–320 nm.

### 3.5. Molecular Docking Studies

The structure of alpha chymotrypsin was obtained from RCSB protein data bank (PDB id: 1YPH) and the ligand structure (AzureC) was downloaded from Pubchem ((https://pubchem.ncbi.nlm.nih.gov/compound/Azure-C, accessed on 14 November 2022). The docking visualization and analysis were carried out using Pymol, Discovery studio, and Autodoc 4.2 program, Autodoc vina. The grid spacing was default set at 0.375 A° with grid size of 40, 40 and 40 A° along with the x, y, and z axis for the analysis.

### 3.6. In Vitro Cytotoxicity Assay

The MTT test was used to examine the human lung adenocarcinoma cells survival after treatment. This test relies on the tetrazolium compound’s capacity to be transformed into an insoluble formazan product in the case of live cells with functioning mitochondria. The cells were seeded in a 96-well plate for 24 h in a CO_2_ incubator to become confluent. The cells were treated with different concentrations of ZnONPs (0.1 mM–5 mM), AzC alone (2.0 µM–12 µM), and AzC-ZnONPs (2.0 µM–12 µM) in quadruplet (*n* = 4) for 24 h. After treatment, the media was discarded, and the cells were incubated with 0.5 mg/mL MTT reagent for three hours. Later, DMSO (100 µL) was added, and the absorbance was recorded at 570 nm with a reference filter of 620 nm using Infinite 200 PRO multimode plate reader (Tecan, Mannedorf, Zurich, Switzerland).

## 4. Results and Discussion

### 4.1. Characterization of AzC-ZnONPs

A cationic dye (AzC) from the phenothiazine family has an anthracene-like aromatic framework with nitrogen and a sulphur atom in the middle ring. On either side of the phenothiazine skeleton in AzC, there is an exocyclic amine group, one of which is secondary amine and the other is primary amine. AzC shows a strong absorption peak at 616 nm in the visible region due to π-π* transition. The well dispersed ZnONPs carry a negative surface charge and many reported studies revealed the interaction of cationic ligand or biomolecule with negatively charged ZnONPs via electrostatic attraction [15]. To determine the size and shape of AzC-ZnONPs after conjugation with AzC, TEM studies were carried out. The TEM images of AzC-ZnONPs, as well as its related histogram plot, are displayed in Figure 1a,b, respectively. The AzC-ZnONPs are spherical in shape, with a mean diameter of 78.39 nm. The conjugation of AzC with ZnONPs was analyzed by hydrodynamic size of the ZnONPs and AzC-ZnONPs using DLS measurement technique, and the results were shown in Figure 1c,d. The average size of the ZnONPs and AzC-ZnONPs was 50.74 nm and 91.28 nm respectively. The increase in size of the AzC-ZnONPs as compared to the ZnONPs alone clearly indicates the conjugation of AzC on ZnONPs surface.

Zeta potential gives the crucial information of surface potential and the colloidal stability of the nanoparticles. The Zeta potential of the ZnONPs and AzC-ZnONPs was calculated and the results are displayed in the Figure 1e,f. The surface potential of AzC-ZnONPs was found to be 10.70 mV and −14.20 mV for ZnONPs respectively. According to the values, the shifting of negative to positive surface charge of ZnONPs to AzC-ZnONPs shows the change in the potential due to the conjugation of AzC on the surface of ZnONPs.

### 4.2. UV-Visible Spectroscopic Studies

UV-Visible titration studies were performed to understand the preliminary binding interaction between the protein and ligand. During the titration, the shifting in absorbance peak of either protein or dyes signifies their interaction or structural alteration [29]. The binding of dye or the nanocomplex to Chymo molecule results in the change in spectra of protein. The UV-Visible spectra showed the characteristic peaks of Chymo, and AzC dye at 280 nm and 617 nm respectively. In this study, the effect was monitored initially by varying the concentration of Chymo and then AzC dye at physiological pH. The spectra of Chymo by varying the concentration of AzC dye show hyperchromic shift along with red shift of 6 nm, which is as shown in Figure 2. The red shift and the isosbestic point in the spectra indicate the complex formation between the AzC dye and Chymo. However, the hyperchromic shift arises due to the change in absorbance at 280 nm, suggesting the conformational change in protein structure [31]. A hyperchromic effect in absorption spectra of Chymo protein arises due to increase in exposure of tryptophan (Trp) residue towards the hydrophilic environment and promotes the binding of AzC with Chymo [32,33]. Moreover, for better clarification of the binding mechanism, in the reverse titration, we varied the amount of Chymo at constant concentration of AzC and it resulted in the hypochromic shift in the absorbance, which indicated the attraction due to the π electron cloud of dye with protein [30].

In another case, the UV-Visible spectroscopic titration was performed in the presence of green synthesized ZnONPs (1.0 mM) to check their effect on the binding of AzC dye and Chymo. The ZnONPs provide surface for the binding of the dye through electrostatic forces like Van der Waal force, hydrogen bonding and hydrophobic interaction etc. To understand the interaction between AzC-ZnONPs and Chymo, spectra were recorded by varying the concentration of Az-ZnONPs complex. This increase in concentration shows a considerable increase in the absorbance of Chymo with a red shift of 10 nm. This demonstrates the strong binding and complex formation between AzC-ZnONPs complex and Chymo. Moreover, in the reverse case, the spectra of AzC-ZnONPs complex were monitored by varying concentrations of Chymo. The increasing concentration of Chymo shows a hypochromic shift with slight red shift. This indicates that ZnONPs enhance the binding capability of AzC with Chymo and AzC-ZnONPs bind strongly as compared to the AzC alone.

For a better understanding and comparison of the binding potential of AzC alone and the AzC-ZnONPs complex, the binding constant was calculated using the Benesi–Hildebrand equation (Equation (1)) [30]
(1)$$\frac{1}{\Delta A}=\frac{1}{\Delta Amax}+\frac{1}{KBH\left(\Delta Amax\right)}\times \frac{1}{\left[M\right]}$$
where Δ*A* is the difference of absorbance maxima of dye alone and bound dye, and [*M*] is the concentration of chymotrypsin. To obtain the binding constant, a linear double reciprocal plot between 1/Δ*A* and 1/[*M*] was plotted (Figure 3).

The assessment of the binding constant values from Table 1 reveals that the AzC-ZnONps complex has high binding potential with Chymo in comparison to the AzC alone. Hence, it concludes that ZnONPs increase the binding affinity of the nanosystem.

### 4.3. Fluorescence Study

Fluorescence spectroscopy is extensively used to understand the interaction, dynamic, and thermodynamic parameters of the ligand’s interaction with biomolecules. Chymotrypsin possesses its natural fluorescence due to the presence of amino acids having fluorophore properties, e.g., tryptophan, tyrosine, and phenylalanine [34,35]. However, the intrinsic fluorescence of Chymo is mainly due to the tryptophan alone, as the fluorescence of tyrosine is quenched in the native structure and the quantum yield of phenylalanine is very low [28,36]. The globular alpha Chymo contains eight tryptophan residues, from which Trp51 and Trp141 are buried in the core, while Trp27 and Trp29 are partially, and Trp172, Trp207 (46%), Trp215, and Trp237 (49%) are fully exposed to the solvent and among them, the latter three Trp units are in direct contact with the active sites [31,37]. Therefore, if any ligand binds close to the Trp residue, then it results in a change in intrinsic fluorescence, which is due to the conformational change, subunit association or denaturation of the protein structure [37]. The fluorescence titration of AzC dye and AzC-ZnONPs with Chymo was performed at two different temperatures (298 K and 323 K) and in the wavelength range of 300–450 nm, as shown in Figure 4. In order to understand the interaction, the AzC was successively added in constant amount of Chymo and the fluorescence emission intensity was recorded at two different temperatures. In another case, an increasing amount of AzC-ZnONPs was titrated with a constant amount of Chymo and the change in the fluorescence spectra was recorded.

The results indicate that AzC binds with Chymo and effectively quenches the fluorescence of Chymo at both temperatures. Whereas, in the case of AzC-ZnONPs complex, a sudden decrease in intensity was observed without any shifting. The binding between them is due to various types of possible molecular interactions like ground state complex formation, collision quenching or energy transfer etc. Quenching in fluorescence intensity indicates the successful binding of AzC with the Trp residues of Chymo and this binding is promoted or accelerated by the presence of nanoparticles (Figure 4). There is no shift observed during titration, which reflects that the dye is in close proximity to the Trp residue and only affects its microenvironment [36].

### 4.4. Mode of Fluorescence Quenching

The intrinsic fluorescence quenching occurs due to the formation of the complex between ligand and protein molecule. The calculated quenching parameters give a quantitative idea of binding and help in a better understanding of the binding mechanism. The fluorescence quenching has two possible mechanisms, static and dynamic quenching, by which the fluorescence of Chymo is effectively quenched. In static quenching, the fluorophore (Trp) and the ligand form a complex at ground state level, whereas in the case of dynamic quenching, the collision occurs between the fluorophore and the ligand [37]. For a better understanding of the quenching mechanism, the well-known Stern–Volmer equation was used, as mentioned below (Equation (2)).
(2)$$\frac{Fo}{F}=1+K_{\mathrm{sv}}[Q]=1+K_{q}\tau_{o}[Q]$$
where *F_0_* and *F* denote the Fluorescence intensity of Chymo in the presence and absence of quencher (AzC and AzC-ZnONPs complex) respectively. K_sv_ is the Stern–Volmer quenching constant, and [Q] is the concentration of the quencher. K_q_ is the biomolecular quenching rate constant, and τ_o_ is the fluorescence average lifetime (2.96 × 10^−9^ s) of the Trp residue of Chymo molecule [31].

To determine the K_sv_ value, a plot between F_0_/F vs. [Q] was plotted and shown in Figure 5a. The linear plot of Stern–Volmer indicates a single type of fluorophore and the slope of the graph gives the value of the Stern–Volmer constant and the quenching rate constant (K_q_), obtained by using the value of K_sv_ and τ_o_ in Equation (3) and summarized in Table 2.
(3)Kq=Ksv/τo

The K_sv_ value gives the idea of the stability of complex formation between the quencher and fluorophore. The higher value of the K_sv_ shows high stability of the complex. The dynamic quenching is based on diffusion and the diffusion constant, which will increase with temperature rise and same will happen with the fluorescence quenching constant. Whereas, in the case of static quenching, the stability of complex decreases with an increase in temperature and shows a decrease in quenching constant [8,16]. From the calculated values of Table 2, it is observed that the K_sv_ value is higher in the presence of ZnONPs, which suggests that the Chymo-AzC-ZnONPs complex is more stable than the Chymo-AzC dye complex. The data show that the value of K_sv_ decreases with an increase in temperature. In addition to this, the value of K_q_ is higher than the maximum dynamic quenching rate constant. These results indicate that the quenching mechanism is governed by static quenching.

### 4.5. Thermodynamic Analysis of Binding

The interaction between the Chymo and AzC/AzC-ZnONPs complex is mainly governed by non-covalent interactions, e.g., Van der Waals interactions, hydrogen bonding, electrostatic forces, and hydrophobic interactions [8]. To understand the nature of binding and the relationship between the protein–ligand interactions, the thermodynamic data play an important role. Various studies reveal that the magnitude and sign of the various thermodynamic parameters, e.g., changes in Gibbs free energy (ΔG), enthalpy (ΔH), and entropy (ΔS), are associated with various interaction forces taking part in binding interactions as mentioned below [8,16]
ΔH > 0 & ΔS > 0 (Hydrophobic Interaction)
ΔH < 0 & ΔS < 0 (Van der Waals force or Hydrogen Binding)
ΔH < 0 & ΔS > 0 (Electrostatic Forces)

The value of thermodynamic parameters between the Chymo and AzC dye, Chymo and AzC-ZnONPs complex were calculated by Van’t Hoff equation and the used thermodynamic equations are given below (Equations (4) and (5)).
(4)$$\ln\left(\frac{K2}{K1}\right)=\frac{\Delta H}{R}\left(\frac{1}{T1}-\frac{1}{T2}\right)$$
Δ*G* = ΔH − TΔS = −RTlnK
(5)

The calculated thermodynamics values are summarized in Table 3.

It can be observed from Table 3 that the values are negative for ΔH, ΔS, and ΔG which means that the predominant force responsible for the interaction is Van der Waals force and hydrogen bonding. Moreover, this interaction process is spontaneous in nature and the high value of ΔH suggests that the whole reaction is enthalpy driven.

### 4.6. Circular Dichroism (CD) Study

CD is a very useful technique to understand the conformational change of DNA and protein structure in interaction with a ligand. The α-Chymo spectrum in CD shows two negative bands, one at 208 nm and other at 230 nm. The change in CD spectra of α-Chymo on interaction with small ligands gives the important information relating to the structural transition from α-helix to β-sheet or in random coiling [37].

In this work, the CD spectra of Chymo titrated with increasing concentrations of AzC and AzC-ZnONPs were recorded (Figure 6). The spectra show that with the addition of AzC dye to the Chymo, the native structure of protein was not preserved. The decrease in molar ellipticity was observed upon the successive addition of AzC dye in Chymo solution, the decrease in molar ellipticity was observed. The difference in the ellipticity of Chymo in the presence and absence of dye shows the partial unfolding of protein. Whereas in the case of AzC-ZnONPs complex, when titrated with the Chymo, spectra show higher change in ellipticity. This reveals that the interaction of dye-NPs complex with Chymo molecule promotes the unfolding of the protein to a much greater extent as compared to the dye alone and shows a significant difference in ellipticity, as shown in Figure 6. The CD spectrum of Chymo in the region of 250–300 nm is dominated by aromatic residues and di-sulphides bridges and from them the contribution is due to tyrosyl L_h_, tryptophanyl L_a_ and L_h_, disulphide n sigma and from them the L_a_ transition is less sensitive than L_b_ for the conformational change [38]. Major CD bands arise due to Phenylalanine (weak band 260 nm), Tyrosine (280 nm), and Tryptophan (300 nm) aromatic amino acids when these aromatic residues move away from each other, leading to decrease in the intensity of CD spectra. The obtained spectra suggest that there is no shifting in peak, and the change in conformation of Chymo structure is greater in case of AzC-ZnONPs as compared to AzC alone, which depicts the stronger binding of AzC-ZnONPs with chymo protein [39].

### 4.7. Molecular Docking

To support the experimental results, molecular docking was performed between chymotrypsin and AzureC. This study gives the idea of the theoretical specific binding site and mode of interaction between the macromolecules (Chymo) and ligand (AzC) and gives the binding energies. To investigate the interaction, the protein PDB file was downloaded from RCSB protein data bank and the ligand (AzC) was taken from PubChem. The docking simulation was performed in Autodock tools 4.2 by taking Chymo molecule as receptor and dye as a ligand. The docking results were visualized in Pymol and Discovery Studio and the possible binding is shown in Figure 7a. Among the various output files, we selected the lowest binding energy structure as the best-docked one. The binding energy from Autodoc Vina came out to be −7.1 kcal/mol in case of AzC. The presence of ZnONPs in the reaction promotes the binding, as supported by the above experimental studies, and enhances its application property. The two-dimensional structure shown in Figure 7b shows that Ala5, Gln116, Ser115, and Phe114 amino acids in AzC play important roles in Van der Waals interaction, hydrogen bonding and hydrophobic interaction. Both fluorescence studies and molecular docking studies suggested that the binding is predominantly by Van der Waals and hydrogen bonding force.

### 4.8. In Vitro Cytotoxicity

A useful approach for determining the therapeutic potential of nano-formulation is the examination of their in vitro cytotoxicity against A549 by examining cellular metabolic activity. The MTT [3-(4,5-dimethylthiazol-2-yl)-2,5-diphenyltetrazolium bromide] reduction test is a colorimetric assay that has been widely used to assess cell viability or cytotoxicity. It is based on metabolically active cells’ NADPH-dependent enzymatic reduction of MTT dye into formazan crystal, which allows for a quantitative assessment of the viability of the cells [8]. The MTT test was utilized in this work to examine and contrast the cytotoxic effects of AzC alone, ZnONPs, and AzC-ZnONPs on human lung adenocarcinoma cell lines (A-549) after 24 h of treatment (Figure 8). The findings reveal that the cytotoxic effects of both AzC and AzC-ZnONPs are dose-dependent and increase with an increase in concentration, whereas in the case of ZnONPs alone, at high concentration there is a dip in toxicity due to agglomeration which reduces the permeability of ZnONPs [40]. From the maximal inhibitory concentration (IC50), the cytotoxicity was found to be 8.2 µM and 6.0 µM for AzC alone and AzC-ZnONPs respectively. The AzC-ZnONPs nanoconjugate show more toxicity compared to the AzC alone at the higher concentration due to the combined ROS generation which results in the killing of more cancer cells as compared to AzC alone [41]. These findings imply that AzC-ZnONPs nanoconjugates should be explored more for the exciting prospect of their use as an anticancer agent.

### 4.9. Biological Relevance

The different types of metallic and non-metallic nanoparticles are used for the delivery purpose of various biomolecules, dyes and drugs for the diagnosis and treatment of various diseases. An evolution in nanoparticle design is required for controlling their behavior, toxicity, and internalization for exploring their more and more use in nanomedicine [12]. The surface modification of nanoparticles encourages the cellular uptake and increases the specificity of nanoparticles. The Chymo corona around the AzC-ZnONPs complex improves its biosafety and biocompatibility and makes it more advantageous for the delivery purpose. The AzC dye with ZnONPs improved its photodynamic therapy (PDT) property and increased its specificity towards the tumor cells [42].

The findings from this paper, obtained through various spectroscopic studies, support the more stable binding of AzC-NPs complex with Chymo as compared to dye alone and the schematic representation of its action is proposed in Figure 9. After entering the tumor cell, the Chymo-AzC-ZnONPs complex might produce reactive oxygen species (ROS) such as singlet oxygen, which can interfere with the lipids and proteins of the organelles and lead to cell apoptosis [43,44,45]. Further, this nanocomplex can be modified to increase its selectivity to distinguish between the healthy and tumor cell. To explore its cell apoptosis mechanism, more in vivo and in vitro studies are required for exploring the use of its PDT property against the tumor cells. Various in vitro studies reported the use of phenothiazinium dye-loaded nanoparticles for its photodynamic property against tumor targeting in various types of cancer [8,46], colon cancer [47], ovarian cancer [48], breast cancer [49], lung cancer [50], gastrointestinal cancer [51], etc. The Chymo-AzC-ZnONPs complex may also be tried as a more effective anti-tumor agent to stop the growth of the cancer cells.

## 5. Conclusions

In this study, we have synthesized AzC-ZnONPs nanoconjugates to explore their effective binding as compared to AzC alone by various physiochemical techniques and molecular docking studies. The results of all the techniques showed a nice correlation and supported each other. The conjugation of AzC with ZnONPs was confirmed by performing DLS, zeta potential before and after conjugation. The TEM and DLS experiments proved that AzC-ZnONPs had a greater mean diameter and hydrodynamic size than ZnONPs. The interaction as well as the complex formation between the Chymo and AzC/AzC-ZnO complex has been investigated by various spectroscopic techniques. The UV-Visible spectroscopy showed hyperchromicity which strongly suggested the complex formation between the AzC-ZnONPs and Chymo molecule. The fluorescence studies and the thermodynamic parameters revealed that the binding was governed by Van der Waals force, hydrogen bonding, and hydrophobic force. The docking study indicated that the AzC gets absorbed on the surface of Chymo molecule and leads to the thermal stability of the complex. According to binding parameters, the AzC-ZnONPs bind with the Chymo molecule as compared to AzC alone, as the presence of ZnONPs promotes the binding and the formation of the complex. Hence, this research suggests a biophysical way to understand the effect of ZnONPs on the protein structure. The green synthesized ZnONPs enhance the stability of the Chymo-AzC-ZnONPs complex and facilitate the delivery of AzC for various anti-cancer therapy. Later, studies on the in vitro cytotoxicity of AzC-ZnONPs against cancer cells clearly showed that they have great anticancer capacity. The experimental results showed that AzC-ZnONPs may be easily synthesized into a biocompatible, affordable nano-formulation that has the potential to cure cancer by combining photo-thermal and chemotherapeutic effects. Hence, we believe that this work might facilitate our understanding regarding the protein–nanoparticle interactions so that such complexes may further be utilized for exploring their biomedical applications for target-specific drug delivery.

## Figures and Tables

**Figure 1 pharmaceutics-15-00074-f001:**
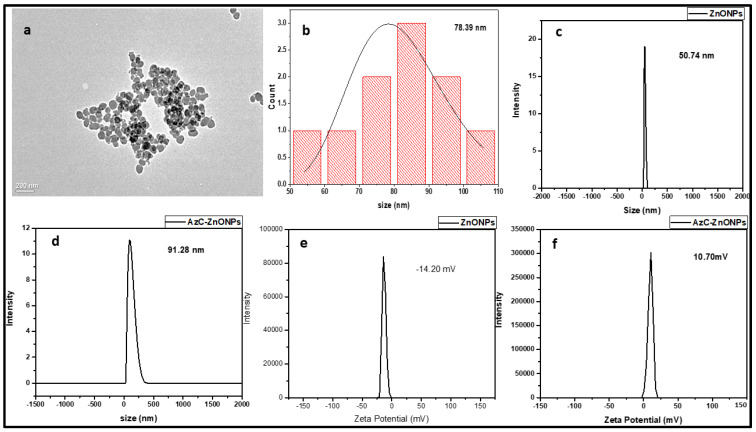
Characterization of AzC-ZnONPs: (**a**,**b**) TEM Image and its corresponding Histogram; DLS Plot of (**c**) ZnONPs, (**d**) AzC-ZnONPs; Zeta potential of (**e**) ZnONPs and (**f**) AzC-ZnONPs.

**Figure 2 pharmaceutics-15-00074-f002:**
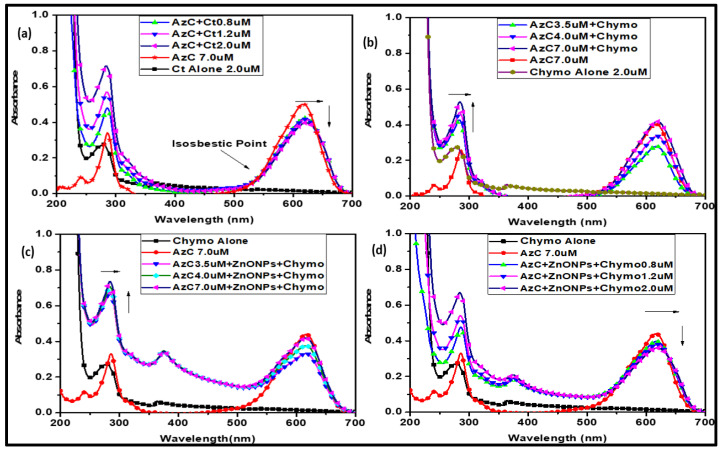
UV-Visible titration Studies: (**a**) Variation of Chymo with constant AzC; (**b**) Variation of AzC with constant Chymo; (**c**) Variation of AzC-ZnONPs with constant Chymo (**d**) Variation of Chymo with constant AzC-ZnONPs.

**Figure 3 pharmaceutics-15-00074-f003:**
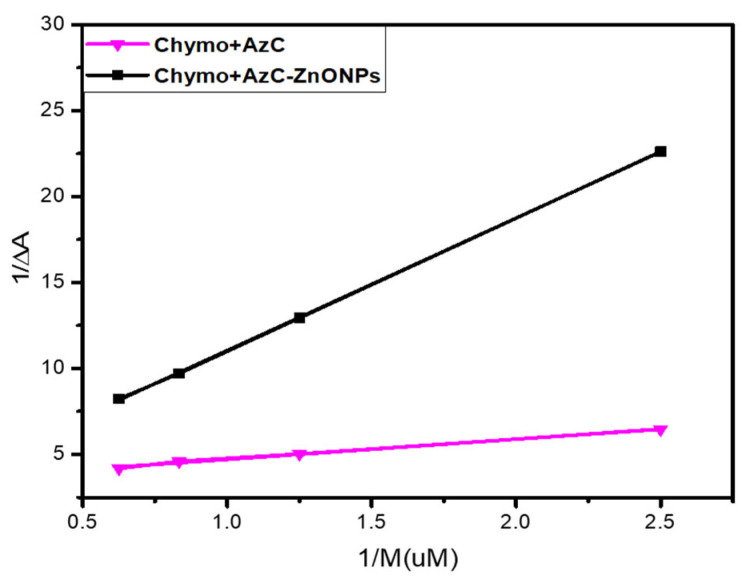
Benesi-Hilderbrand Plot of Chymo+AzC alone and Chymo+AzC-ZnONPs complex.

**Figure 4 pharmaceutics-15-00074-f004:**
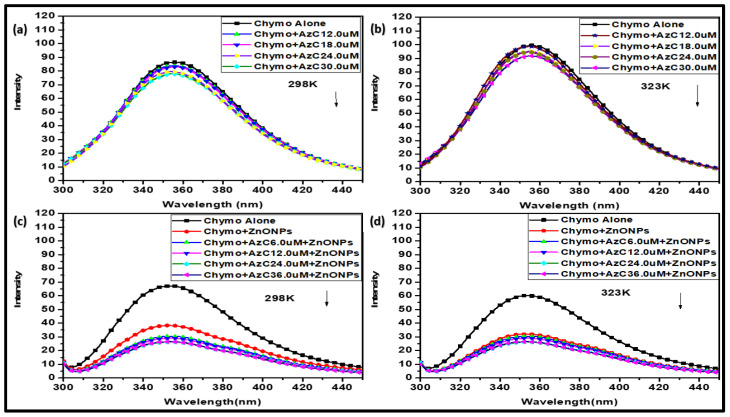
Fluorescence spectra of Chymo in presence of AzC at 298 K (**a**) & 323 K (**b**); and AzC-ZnONPs complex at 298 K (**c**) & 323 K (**d**).

**Figure 5 pharmaceutics-15-00074-f005:**
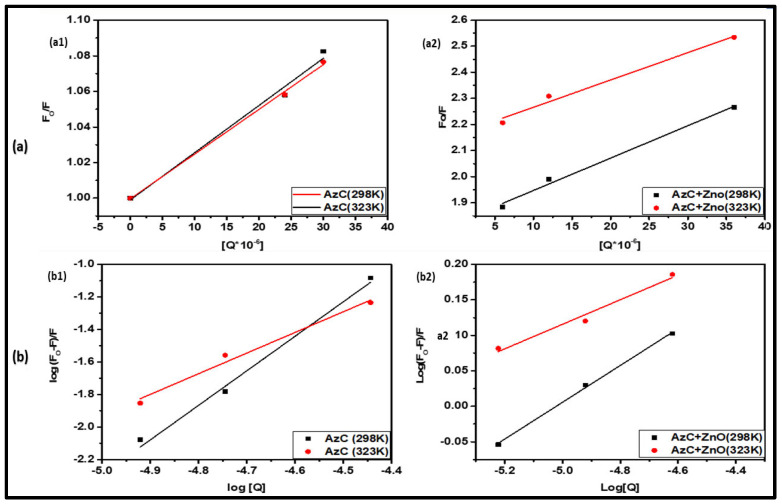
(**a**) Stern-Volmer plot of Chymo-AzC (**a1**) and AzC-ZnONPs (**a2**) at two different temperatures 293 K and 323 K. (**b**) Double logarithm plot for fluorescence quenching of Chymo-AzC (**b1**) and AzC-ZnONPs (**b2**) at temperatures 293 K and 323 K.

**Figure 6 pharmaceutics-15-00074-f006:**
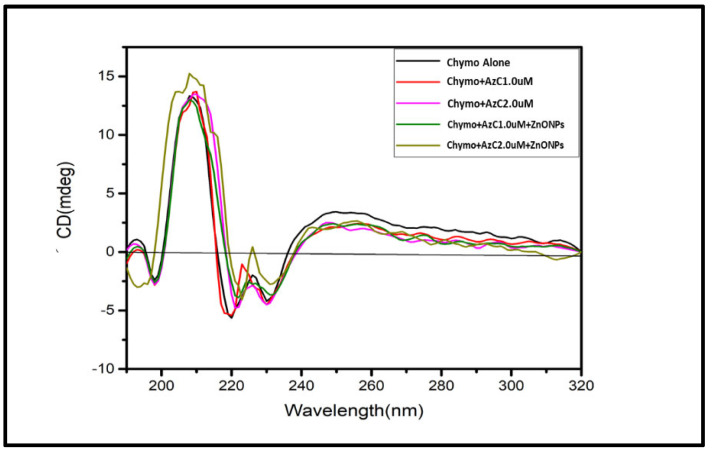
CD spectra of Chymo in the presence of AzC and AzC-ZnONPs complex.

**Figure 7 pharmaceutics-15-00074-f007:**
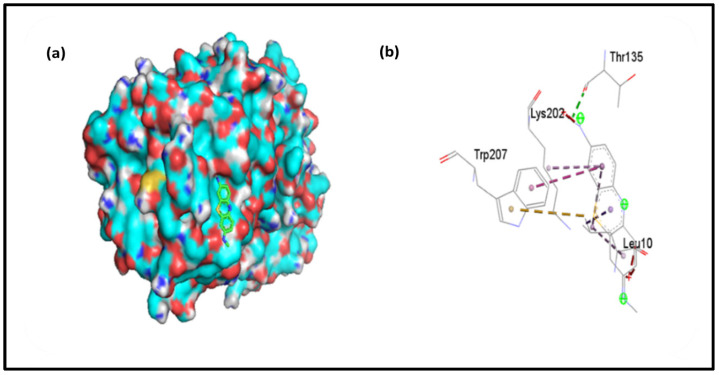
(**a**) 3D representation of Binding of AzC with Chymo, (**b**) 2D picture of AzC with Chymotrypsin.

**Figure 8 pharmaceutics-15-00074-f008:**
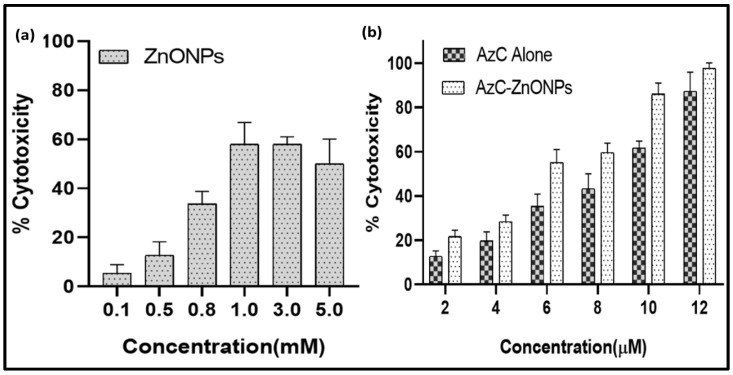
Cytotoxicity study of A-549 cell treated with different concentrations of (**a**) ZnONPs; (**b**) AzC alone and AzC-ZnONPs (*n* = 4).

**Figure 9 pharmaceutics-15-00074-f009:**
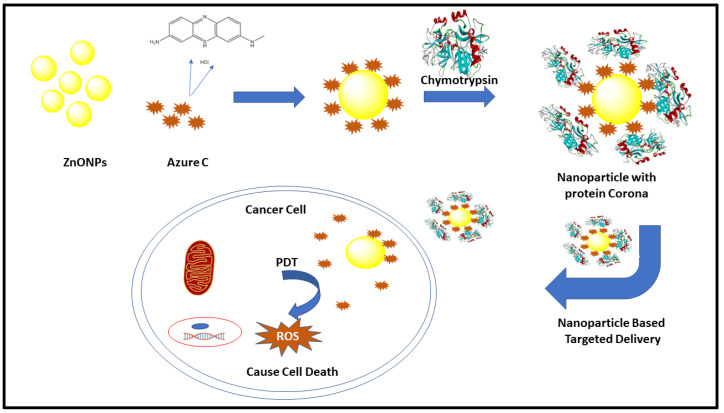
Biological relevance of nanoparticle-based targeted delivery against cancer.

**Table 1 pharmaceutics-15-00074-t001:** Variation in binding constant of Chymo with AzC and AzC-ZnONPs complex.

S. No.	Complex	Binding Constant
1,	Chymo+AzC	5.72 × 10^4^ M^−1^
2,	Chymo+AzC-ZnONPs	11.49 × 10^5^ M^−1^

**Table 2 pharmaceutics-15-00074-t002:** The calculated values of Stern-Volmer constant (K_SV_), quenching constant (Kq) of Chymo+AzC and AzC-ZnONPs at two different temperatures 293 K and 323 K.

S.No	Complex (Temperature)	Ksv (mole^−1^)	Kq (Quenching Rate Constant) (M^−1^ s ^−1^)	R^2^
1,	Chymo+AzC (298 K)	2.6 × 10^3^	8.78 × 10^11^	0.991
2,	Chymo+AzC (323 K)	2.5 × 10^3^	8.44 × 10^11^	0.998
3,	Chymo+AzC-ZnONPs (298 K)	12.4 × 10^3^	4.18 × 10^11^	0.985
4,	Chymo+AzC-ZnONPs (323 K)	10.4 × 10^3^	3.51 × 10^11^	0.993

**Table 3 pharmaceutics-15-00074-t003:** Thermodynamic Parameters for the interaction of Chymo-AzC and AzC-ZnONPs at two different temperatures (298 K & 323 K).

Complex	ΔH (Kjmol^−1^)	ΔG (Kjmol^−1^)	ΔS (Kjmol^−1^ K^−1^)
Chymo+AzC (298 K)	−0.354	−7.434	−2.19 × 10^−2^
Chymo+AzC (323 K)	−0.354	−5.823	−1.83 × 10^−2^
Chymo+AzC-ZnONPs (298 K)	−281.666	−47.489	−78.58 × 10^−2^
Chymo+AzC-ZnONPs (323 K)	−281.666	−22.364	−80.27 × 10^−2^

## Data Availability

All data published herein will be made available upon request.

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
