# Peer review of "Influence of Green Synthesized Zinc Oxide Nanoparticles on Molecular Interaction and Comparative Binding of Azure Dye with Chymotrypsin: Novel Nano-Conjugate for Cancer Phototherapy"

_pharmaceutics, 2022, doi:10.3390/pharmaceutics15010074_

Round 1

Reviewer 1 Report

In this study, the author team synthesized organic/inorganic hybrid nanosystems consisting of AzureC (photosensitive dye) conjugated with rennet protease protein (Chymo) and zinc oxide nanoparticles (ZnONPs) and found that it can be used as a novel nanocoupling for phototherapy of various cancers. It is interesting to construct novel nanocouples with green synthesis techniques. However, there are some problems with this manuscript. My comments are as follows.

Main points.

1. the first sentence of the abstract mentions that combination therapy is considered an effective strategy for the treatment of serious diseases, including cancer-type diseases. It is suggested that it would be more appropriate to put it after talking about the shortcomings of conventional treatment before talking about the advantages of combination therapy.

2. In the abstract it is mentioned that zinc oxide nanoparticles have interesting zinc oxide nanoparticles have interesting physicochemical and anticancer properties, it can be briefly described what are the properties that we focus on.

3. In the abstract AzureC is not written in the same way as in the keyword Azure C. It is suggested to revise and add the abbreviation (AzC).

4. the first two keywords are repeated and there are more keywords, suggest streamlining.

5. The first paragraph of the preface is about epidemiology and lacks statistical results of data in recent years.

6. The last two sentences of the third paragraph of the preface are missing references.

7. 4.2 The first paragraph of the literature is arranged in the wrong order.

8. 4.2 The case of ZnONPS (1mM) in the second paragraph is not consistent with the figure.

9. The same name in Fig. 2 and Fig. 4 is suggested to be labeled with a uniform color, e.g. AzC 7.0uM.

10. When the authors gave examples of the photodynamic properties of phenothiazine dye-loaded nanoparticles for various types of cancer, the examples given were early. It is recommended to re-find examples from the last 3 years.

Secondary points.

1. the notes of the vertical coordinate plot in Figure 5a are misplaced.

2. 298 and 323 units in entries 3 and 4 in Table 2 are not indicated

3. references are not indicated in the contents of 4.5

4. references in the material section are cited in the wrong format.

5. There are some minor grammatical errors in the text, which are suggested to be corrected.

6. In the citation literature formatting errors, letters appear in the year, such as references 29 and 34.

Author Response

  1. the first sentence of the abstract mentions that combination therapy is considered an effective strategy for the treatment of serious diseases, including cancer-type diseases. It is suggested that it would be more appropriate to put it after talking about the shortcomings of conventional treatment before talking about the advantages of combination therapy.

Response: As suggested by the reviewer, we have now done the changes in the abstract and the changes are highlighted in the manuscript, as follows:

Till date, different types of conventional drugs have been used to fight tumors. However, they have significant flaws, including their usage being constrained because of their low bioavailability, poor supply, and serious side effects. The modern combination therapy has been viewed as a potent strategy for treating serious illnesses including cancer-type feared diseases. The nanoparticles are a promising choice for cancer therapeutic and diagnostic applications because of their fascinating optoelectronic and physicochemical features. Among the metallic nanoparticles, Zinc oxide nanoparticles possess interesting physicochemical and anticancer characteristics such as ROS generation, high retention, enhanced permeability etc. that make these  attractive candidates for the treatment and diagnosis of cancer. Zinc oxide nanoparticles showed toxicity via excessive reactive oxygen species (ROS) production, and by the destruction of mitochondrial membrane. Here, we have synthesized organic/inorganic hybrid nanosystem composed of chymotrypsin protein (Chymo) with AzureC (AzC) conjugated with zinc oxide nanoparticles (ZnONPs). The conjugation of AzureC with ZnONPs was confirmed by transmission electron microscopy (TEM), Zeta potential and Dynamic light scattering (DLS) experiment. The interaction of Chymo with AzC alone and AzC-ZnONPs was investigated and it was observed that the interaction was enhanced in the presence of ZnONPs, which was concluded by the results obtained from different spectroscopic techniques such as UV-Visible spectroscopy, fluorescence spectroscopy and circular dichroism in combination with molecular docking. UV-Visible spectroscopic studies and the corresponding binding parameters showed that the binding of AzC-ZnONPs complex with Chymo is much higher than that of AzC alone. Also, the fluorescence measurement shows enhancement in static quenching during titration of Chymo with AzC-ZnONPs as compared to dye alone. In addition to this, circular dichroism results show that the dye and dye-NPs conjugate do not cause much structural change in α-Chymo. The molecular docking and thermodynamic studies show the predominance of hydrogen bonding, Van der Waal force and hydrophobic forces during the interactions. After correlation of all the data, interaction of Chymo with AzC-ZnONPs complex shows strong interaction as compared to dye alone. The moderate binding with chymo without any alteration in the structure makes it desirable for the distribution and pharmacokinetics. Also, in vitro cytotoxicity of the AzC-ZnONPs demonstrated on A-549 adenocarcinoma cell lines. Our findings from physiochemical investigations suggest that the chymotrypsin coated AzC conjugated ZnONPs could be used as a novel nanoconjugate for the various cancer phototherapies.

  1. In the abstract it is mentioned that zinc oxide nanoparticles have interesting zinc oxide nanoparticles have interesting physicochemical and anticancer properties, it can be briefly described what are the properties that we focus on. 

Response: As suggested by the learned referee, we have now done the changes in the abstract. Kindly refer to the highlighted portion of the response to comment 1, as above.

  1. In the abstract AzureC is not written in the same way as in the keyword Azure C. It is suggested to revise and add the abbreviation (AzC). 

Response: As suggested the correction has been done.

  1. the first two keywords are repeated and there are more keywords, suggest streamlining.

Response: Correction done

Keywords: Nanoparticles, Zinc oxide nanoparticles; A-549 cell lines; AzureC; Chymotrypsin; Cancer phototherapy; Dye-ZnO complex; Protein nanoparticle interaction

  1. The first paragraph of the preface is about epidemiology and lacks statistical results of data in recent years. 

Response: As per learned referee, the statistical data is now added in the first paragraph of the introduction and is highlighted as below:

Cancer continues to be one of the leading causes of mortality worldwide. According to statistics, more than 19 million cases of cancer were reported throughout the world till September 2022 [1], with estimated 10 million deaths by 2020 [1,2]. The scientific community is continuously working to create new diagnostic tools and therapeutic approaches to deal with this deadly disease.

6 The last two sentences of the third paragraph of the preface are missing references.

Response: Kindly accept our apologies for this error and we truly appreciate the critical observation of the learned reviewer. We have now added the references and highlighted them in the revised manuscript.

  1. 4.2 The first paragraph of the literature is arranged in the wrong order. 

Response: As suggested, we have now rearranged the text and highlighted in the revised manuscript

  1. 4.2 The case of ZnONPS (1mM) in the second paragraph is not consistent with the figure.

Response: Many thanks for this critical observation. Correction has been done accordingly.

  1. The same name in Fig. 2 and Fig. 4 is suggested to be labeled with a uniform color, e.g. AzC 7.0uM. 

Response: Thanks again for the critical observation. Figures have now been labeled accordingly.

  1. When the authors gave examples of the photodynamic properties of phenothiazine dye-loaded nanoparticles for various types of cancer, the examples given were early. It is recommended to re-find examples from the last 3 years. 

Response: As suggested by the learned referee, we have now added few recent references of the photodynamic properties of phenothiazine dye-loaded nanoparticles for various types of cancer. The added references are as follows;

  • Sonia, Singh, A., Shivangi, Kukreti, R., Kukreti, S., & Kaushik, M. (2022). Probing multifunctional azure B conjugated gold nanoparticles with serum protein binding properties for trimodal photothermal, photodynamic, and chemo therapy: Biophysical and photophysical investigations. Materials Science and Engineering: C, 112678.
  • Nagi, J. S., Skorenko, K., Bernier, W., Jones, W. E., & Doiron, A. L. (2019). Near infrared-activated dye-linked ZnO nanoparticles release reactive oxygen species for potential use in photodynamic therapy. Materials, 13(1), 17.
  • Vilsinski, B. H., Gonçalves, R. S., Caetano, W., Souza, P. R. D., Oliveira, A. C. D., Gomes, Y. S., ... & Muniz, E. C. (2021). Photodynamic Therapy: Use of Nanocarrier Systems to Improve Its Effectiveness. In Functional Properties of Advanced Engineering Materials and Biomolecules (pp. 289-316). Springer, Cham.
  • Rajan, D., & Ilanchelian, M. (2018). Exploring the interaction of Azure dyes with t-RNA by hybrid spectroscopic and computational approaches and its applications toward human lung cancer cell line. International journal of biological macromolecules, 113, 1052-1061.
  • Avgustinovich, A. V., Bakina, O. V., Afanas' ev, S. G., Cheremisina, O. V., Spirina, L. V., Dobrodeev, A. Y., ... & Choynzonov, E. L. (2021). Nanoparticles in Gastric Cancer Management. Current Pharmaceutical Design, 27(21), 2436-2444.
  • Kim, S., Lee, S. Y., & Cho, H. J. (2018). Berberine and zinc oxide-based nanoparticles for the chemo-photothermal therapy of lung adenocarcinoma. Biochemical and biophysical research communications, 501(3), 765-770.

Secondary points. 

1.the notes of the vertical coordinate plot in Figure 5a are misplaced. 

Response: Correction done

2.298 and 323 units in entries 3 and 4 in Table 2 are not indicated 

Response: Correction done

  1. 3. references are not indicated in the contents of 4.5 

Response: Kindly accept our apologies for this error. We have added the references in the section 4.5

  1. 4. references in format. material section is cited in the wrong 

Response: Correction done

5.There are some minor grammatical errors in the text, which are suggested to be corrected. 

Response: As suggested by the learned reviewer, we have thoroughly checked the

manuscript to correct the grammatical errors and spelling errors.

6.In the citation literature formatting errors, letters appear in the year, such as references 29 and 34. 

Response: Correction done

Reviewer 2 Report

The authors have presented interesting work on zinc oxide nanoparticles on molecular interaction on the comparative binding of azure dye with chymotrypsin. This research may be useful in cancer phototherapy. The introduction is nicely presented. The importance of research is highlighted. The experimental part is presented in detail. The conclusions are based on obtained results. Still, the work is not presented according to the instructions for the authors:

1. letter format for keywords

2. Figures should be presented in the same manner: the authors use letters A, a, and a. for figures (Figure 1a)...; Figure 2A) ....; Figure 5a.) ...)

3. equations are not justified

4. Figure 3. capture: space should be removed

5. Line 369: space should be removed

6. Tables: samples should be labeled as numbers 1, 2, 3, rather than 1., 2., 3., .... 

7. List of references: it should be prepared according to instructions, also letter type, format, and space should be corrected. 

Author Response

      1.letter format for keywords 

Response: As suggested by the reviewer, we have now corrected the keywords in the manuscript. The changes are as follows:

Nanoparticles, Zinc oxide nanoparticles; A-549 cell lines; AzureC; Chymotrypsin; Cancer phototherapy; Dye-ZnO complex; Protein nanoparticle interaction

2.Figures should be presented in the same manner: the authors use letters A, a, and a. for figures (Figure 1a).; Figure 2A). Figure 5a.)

Response: Kindly accept our apologies for this error. We truly appreciate the critical observation of the learned reviewer. Corrections have now been done in the figures and their legends accordingly.

3.equations are not justified 

Response: As suggested, we have corrected the formatting of all the equations in the manuscript.

4.Figure 3. capture: space should be removed 

Response: Correction done

5.Line 369: space should be removed

Response: Correction done

6.Tables: samples should be labeled as numbers 1, 2, 3, rather than 1., 2., 3., 

Response: Correction done 

  1. List of references: it should be prepared according to instructions, also letter type, format, and space should be corrected.

      Response: Correction done

Reviewer 3 Report

This manuscript shows the study to increase the effect of photodynamic therapy by using AzureC and chymo in zinc oxide. However, the results of In-vitro experiments using cancer cells or Iv-vivo systems using animals are not presented in the manuscript. In order for this manuscript to be published in the pharmaceutics journal, it must be applied to an actual model to prove the increased cancer therapy effect.

Author Response

This manuscript shows the study to increase the effect of photodynamic therapy by using AzureC and chymo in zinc oxide. However, the results of In-vitro experiments using cancer cells or Iv-vivo systems using animals are not presented in the manuscript. In order for this manuscript to be published in the pharmaceutics journal, it must be applied to an actual model to prove the increased cancer therapy effect.

Response: As suggested by learned reviewer, we have now performed a new MTT experiment (Figure 8 of revised manuscript) to determine the cytotoxicity of AzC dye, ZnONPs and AzC-ZnONPs against Human lung adenocarcinoma cell lines (A-549). The experimental method details and results are added and highlighted as section 2, section 3.6 and section 4.8 of revised MS.

The added text as section 4.8 having the discussion of figure 8 in revised MS is as follows:

4.8 In Vitro Cytotoxicity

A useful approach for determining the therapeutic potential of nano-formulation is the examination of their in vitro cytotoxicity against A549 lung cancer cell line by examining cellular metabolic activity. The MTT [3-(4,5-dimethylthiazol-2-yl)-2,5-diphenyltetrazolium bromide] reduction test is a colorimetric assay that has been widely used to assess cell viability or cytotoxicity. It is based on metabolically active cells' NADPH-dependent enzymatic reduction of MTT dye into formazan crystal, which allows for a quantitative assessment of the viability of the cells [8]. The MTT test was utilized in this work to examine & contrast the cytotoxic effects of AzC alone, ZnONPs and AzC-ZnONPs on A549 cell lines after 24 hours of treatment (Fig. 8). The findings reveal that the cytotoxic effects of both AzC and AzC-ZnONPs are dose-dependent and increase with an increase in concentration. However, in case of ZnONPs alone, at high concentration, there is a dip in toxicity due to agglomeration which reduces the permeability of ZnONPs [40]. From the maximal inhibitory concentration (IC50), the cytotoxicity was found to be 8.2 µM and 6.0 µM for AzC alone and AzC-ZnONPs respectively.  The AzC-ZnONPs nanoconjugate showed more toxicity compared to the AzC alone at the higher concentration due to the combined ROS generation, which results in the killing of more cancer cells as compared to AzC alone [41]. These findings imply that AzC-ZnONPs nanoconjugates should be explored more for the exciting prospect of their use as an anticancer agent.

Figure 8: Cytotoxicity study of A-549 cell treated with different concentrations of a) ZnONPs; b) AzC alone and AzC-ZnONPs (n=4)

Round 2

Reviewer 1 Report

The author has given clear answers to all my questions. I suggest accept.

Reviewer 3 Report

The queries have been sufficiently addressed.